# Expression of Bitter Taste Receptors in the Intestinal Cells of Non-Human Primates

**DOI:** 10.3390/ijms21030902

**Published:** 2020-01-30

**Authors:** Hiroo Imai, Miho Hakukawa, Misa Hayashi, Ken Iwatsuki, Katsuyoshi Masuda

**Affiliations:** 1Molecular Biology Section, Department of Cellular and Molecular Biology, Primate Research Institute, Kyoto University, Aichi 4848506, Japan; akukawa.miho.3e@kyoto-u.ac.jp (M.H.); hayashi.misa.77s@st.kyoto-u.ac.jp (M.H.); 2Department of Nutritional Science and Food Safety, Faculty of Applied Bioscience, Tokyo University of Agriculture, Tokyo 1568502, Japan; ki204886@nodai.ac.jp; 3Structural Bioscience for Taste Molecular Recognition, Graduate School of Medicine, Kyoto University, Kyoto 6068507, Japan

**Keywords:** bitter taste receptors, RNAseq, macaque

## Abstract

(1) Background: Recent studies have investigated the expression of taste-related genes in the organs of various animals, including humans; however, data for additional taxa are needed to facilitate comparative analyses within and among species. (2) Methods: We investigated the expression of taste-related genes in the intestines of rhesus macaques, the non-human primates most commonly used in experimental models. (3) Results: Based on RNAseq and qRT-PCR, genes encoding bitter taste receptors and the G-protein gustducin were expressed in the gut of rhesus macaques. RNAscope analysis showed that one of the bitter receptors, TAS2R38, was expressed in some cells in the small intestine, and immunohistochemical analysis revealed the presence of T2R38-positive cells in the villi of the intestines. (4) Conclusions: These results suggest that bitter receptors are expressed in the gut of rhesus macaques, supporting the use of macaques as a model for studies of human taste, including gut analyses.

## 1. Introduction

Taste perception can provide important information about foods. In mammals, bitter taste in the oral cavity is detected by T2Rs, which are heterotrimeric G-protein-coupled receptors expressed on the surface of the tongue [1]. When bitter compounds bind to receptors, information is transmitted to the brain, enabling the recognition of bitter taste.

Non-human primates such as macaques and marmosets, which belong to the Old World and New World monkey families, respectively, are frequently used as models for humans in various types of research. Previous studies have shown that taste-related proteins (e.g., gustducin, TRPM5, and some T2Rs) are selectively expressed in the cecum of common marmosets [2,3]. These observations suggest that T2R expression patterns vary among species, even among primates. In addition, the functional properties of T2Rs are different even among orthologous species. For example, T2R38 in humans [4] and macaques [5] is responsive to phenylthiocarbamide (PTC) but the mouse orthologue T2R138 is not [6]. In humans, a polymorphism in the TAS2R38 gives rise to the difference in response to PTC between the PAV (taster) and AVI (non-taster) genotypes (phenotypes) [4]. Also, in Japanese macaques, there is a polymorphism in the TAS2R38 gene in which a mutation in the nucleotide position 2 of the starting methionine results in the non-taster phenotype [5].

A previous study using mouse and human organs suggested that T2R expression patterns vary among cell types. For example, Tas2R131 is expressed in Paneth cells and goblet cells in the mouse small and large intestines, respectively [7], T2R138 is expressed in the enteroendocrine cells of the mouse colon [8], and T2R38 is expressed in the enteroendocrine cells of the human colon [9]. To understand the general mechanism by which T2R expression is regulated in humans and rodents, it is necessary to use an experimental model that is intermediate between these species. In this study, we measured TAS2R mRNA levels and corresponding protein expression levels in rhesus macaques. This non-human primate could be a suitable model for studying human taste perception by focusing in particular on the TAS2R38 gene and T2R38 protein for which appropriate antibodies are readily available.

## 2. Results

### 2.1. RNAseq and RT-PCR

We first used RNAseq to screen for TAS2R mRNAs in various organs of rhesus macaques, which have orthologues to human TAS2Rs (Appendix A and Figure 1). 

We found that some TAS2Rs are highly expressed in the rhesus macaque gut. In particular, TAS2R1, 3, 4, 5, and 46 were expressed in nearly all gut samples, while TAS2R14, 19, 20, and 38 were expressed in only certain tissues. RNAseq failed to detect other TAS2Rs. Of note, TAS2R38 expression in the intestine is related to the risk of intestinal cancers associated with the taster/non-taster phenotype [10]. To confirm the expression pattern of TAS2R38 in the intestine, we performed RT-qPCR analyses of TAS2R38, GNAT3 (gustducin), and TRPM5 (Tables S2, S3 and Figure 2). TAS2R38 and GNAT3 were both highly expressed in the circumvallate papillae (CV), but TAS2R38 was moderately expressed in the ileum and duodenum as compared with GNAT3 [3]. Meanwhile, TRPM5 was highly expressed in the upper digestive system. These results revealed an expression pattern similar to that in human tissues.

### 2.2. RNAscope

Next, we evaluated tissue localization of TAS2R38 by RNAscope and immunohistochemistry. RNAscope detects mRNAs by in situ hybridization. As previously described, RNAscope detected TAS2R38 and TRPM5 in taste receptor cells (Figure 3A). We detected weak positive signals in gut tissues (Figure 3B–D). 

### 2.3. Immunohistochemistry

We evaluated T2R38 signals by immunohistochemistry (Appendix A). Using T2R38 antibodies, we detected signals in the CV, and these signals overlapped with GNAT3 signals (Appendix A). To confirm antibody reactivity, we used HEK293 cells transiently transfected with rhesus macaque TAS2R38 cDNA. Transfected cells were then stained using each antibody (Appendix A). As shown in Appendix A, T2R38 protein was detected with response to PTC. We confirmed that each antibody could specifically detect T2R38 protein in rhesus macaques. We further examined T2R expression in cells of the intestinal villi in the duodenum that were positive for T2R38. There are several types of cells in the intestinal epithelium and expression of T2Rs is limited to specific types: Paneth cells in the small intestine, [7] goblet cells in the large intestine [11], enteroendocrine cells [9], and tuft cells [12]. T2R38 antibody immunoreactivity was observed in both enteroendocrine cells and goblet cells marked by chromogranin A (arrowheads in Figure 4A–C) [13] and mucin2 (arrowheads in Figure 4D–F) [14], respectively. We also detected co-expression of GNAT3 and T2R38 (arrowheads in Figure 4J-L), making GNAT3-dependent signal transduction in these cells possible. We have previously observed that rhesus macaque T2R38 can activate G-protein with the C-terminus of GNAT3 (G16/gust44) [15] in a heterologous expression system [5], so this may also be possible in intestinal cells. Several DCLK1-positive tuft cells were negative for T2R38 (arrows in Figure 4H,I); however, some cells were double-stained by T2R38 and DCLK1 antibodies (Arrowheads in Figure 4G-I). We did not obtain evidence of T2R38 expression in Paneth cells, possibly due to the functionality of the Paneth cell marker antibody. Similar expression profiles have been observed in the ileum (Appendix A) and colon (Appendix A) of rhesus macaques, suggesting differences from mice. In mice, the Tas2R131 promoter functioned in Paneth cells in the small intestine [7] and in goblet cells in the large intestine [11]. These results indicate that bitter taste receptors, or at least T2R38, are expressed in the intestines of rhesus macaques, as in humans, and have similar functionality. Accordingly, rhesus macaques are suitable models for human research, owing to these similarities in receptivity and expression profiles, at least for T2R38 receptors.

## 3. Discussion

### 3.1. Expression Profiles

We screened the expression profiles of TAS2Rs and related molecules in the gut of rhesus macaques. RNAseq revealed that TAS2R1, 3, 4, 5, 20, and 46 were expressed in the duodenum, ileum, and colon, consistent with observations using human cultured cell lines [16] and the RefEX and GTEx databases; the exception is TAS2R1, which was not expressed in humans but was substantially expressed in rhesus macaque tissues. The expression of TAS2R119, an orthologue of primate TAS2R1 in mice, has been reported in the mouse duodenum, jejunum, ileum, and colon [7], suggesting that TAS2R1 exhibits similar expression patterns to TAS2R119 in mice. However, the expression patterns of other genes were very similar to those in humans, suggesting that rhesus macaque is a good model for the human gut. In particular, the expression of TAS2R38 in rhesus macaques was similar to that in humans, with higher levels in the small intestine than in the large intestine, but was not similar to the expression of the mouse orthologue Tas2R138, which is more highly expressed in the large intestine than in the small intestine [7]. The expression patterns of TAS2R3, 4, and 5 were similar among species, while TAS2R5 expression was lost in mice. Owing to the numerous mouse orthologues of human and rhesus macaque TAS2R14 as well as the high number of orthologues of mouse TAS2R46 in humans and rhesus macaques, it is difficult to directly compare the expression levels of these genes between rodents and primates. 

### 3.2. TAS2R-Positive Cells

We observed the expression of TAS2R38 with gustducin in enteroendocrine cells and goblet cells in the rhesus macaque intestine. These results are basically consistent with those of previous studies in mice and humans; mouse T2R131 is expressed in Paneth cells in the small intestine and in goblet cells in the large intestine [7,11]. T2R138, an orthologue of primate T2R38, is expressed in the enteroendocrine cells of the mouse ileum and colon [8]. T2R38 is also expressed in the enteroendocrine cells of the human colon [9]. We found that T2R38 is expressed in DCLK1-positive cells in the intestine. DCLK1 is a marker for tufted cells (reviewed in [17]). In mice, the expression of T2Rs in tuft cells is associated with parasitic infection [12]. T2Rs include T2R138, consistent with our data in rhesus macaques. Therefore, the basic expression patterns are similar to those in other species, although it is necessary to evaluate the expression patterns of each type of T2R. Our results indicate that rhesus macaques would be an appropriate model for analyses of specific phenotypes in the gut.

## 4. Materials and Methods

### 4.1. Materials

Samples of rhesus macaque gut were acquired from the stored tissue bank (consisting of tissues obtained from dead rhesus macaques) at the Primate Research Institute, Kyoto University (Permit Numbers: 2016-054 (1 April 2016); 2017-016 (1 April 2017); 2018-004 (1 April 2018)). For the gene expression analysis, we used tissues from six rhesus macaques. Small blocks of tissue were incubated at 4 °C in RNAlater (Agilent Technologies, Santa Clara, CA, USA) overnight and stored at −80 °C until use. For immunohistochemical analysis, we used tissues from three rhesus macaques. Small blocks of tissue were incubated in 4% paraformaldehyde at 4 °C before use. Additional chemicals were purchased from commercial suppliers. 

### 4.2. RNAseq and RT-PCR

Total RNA was extracted from tissues using an RNeasy Plus Mini Kit (Qiagen GmbH, Hilden, Germany) and its quality was confirmed using a Bioanalyzer (Agilent Technologies). Genomic DNA removal and reverse transcription were performed using the PrimeScript RT Reagent Kit with gDNA Eraser (Takara, Kyoto, Japan) for 1 μg of total RNA with and without enzymes. RNAseq was performed at The Research Institute for Microbial Diseases, Osaka University, using a TruSeq Stranded mRNA Sample Prep Kit and HiSeq2500 as the sequencer. Analysis was performed by GeneBay to confirm the annotation of TAS2Rs of RheMac2, as shown in Appendix A. Real-time PCR was performed using the StepOne Plus system (Applied Biosystems, Foster City, CA, USA) with Thunderbird SYBR qPCR Mix (Toyobo, Tokyo, Japan) and the primers listed in Appendix A. Standard conditions (95 °C for 30 s, 50 cycles of 95 °C for 15 s, and 60 °C for 45 s) were applied and analyses were performed in triplicate. Quantities of mRNA for TAS2R38, GNAT3 (gustducin), and TRPM5 were determined relative to the quantities of the internal markers GAPDH and β-actin [18,19]. Amplicons were also analyzed by 2% agarose gel electrophoresis and the presence of single bands was confirmed.

### 4.3. RNAscope

RNAscope experiments were conducted according to the manufacturer’s instructions (Advanced Cell Diagnostics, Newark, CA). For the manual single-plex RNAscope assays, tissue sections were baked for 1 h at 60 °C. Deparaffinized treatment was performed at room temperature. Target retrieval was performed for 15 min at 100 °C, followed by protease treatment for 30 min at 40 °C. Probes were then hybridized for 2 h at 40 °C. RNAscope amplification was performed by DAB chromogenic detection. Images were acquired by all-in-one microscopy (BZ-X710; Keyence, Osaka, Japan).

### 4.4. Immunohistochemistry

Stored samples were embedded in paraffin and sectioned at 10 μm. Sections were deparaffinized and rehydrated in a graded alcohol series followed by water. To unmask immunogenic epitopes, antigen retrieval was performed with RNAscope target retrieval reagents (Advanced Cell Diagnostics) diluted to 1:10 in distilled water at 98–100 °C for 15 min. Then, an antigen epitope was fixed in 100% ethanol for 5 min. Nonspecific binding was blocked with 10% normal horse serum (S-2000; Vector Laboratories, Burlingame, CA, USA) diluted in 0.01% Tween-20/0.01 M phosphate-buffered saline (PBS-T) and incubated at room temperature for 1 h. Sections were incubated overnight at 4 °C with primary antibodies (Appendix A) diluted in blocking solution (10% normal horse serum/PBS-T). Sections were then incubated with secondary antibodies (Appendix A) diluted in blocking solution at room temperature for 1 h. Nuclei were stained by a 4’,6-Diamidino-2-phenylindole (DAPI) solution (Dojindo Molecular Technologies, Tokyo, Japan) diluted at 1:1000 in PBS-T at room temperature for 1 min. Sections were then washed and embedded by Mountant Permafluor (TA-030-FM; Thermo Fisher Scientific, Waltham, MA, USA). Fluorescent images were acquired by a fluorescence microscope (LSM-510, Axioplan 2; Carl Zeiss Microscopy GmbH, Jena, Germany) mounted on a cooled charge-coupled device camera system (Cool SNAP HQ; Photometrics, Tucson, AZ, USA). Image acquisition and processing were performed using an Apple Macintosh Power Mac G4 computer running IPLab spectrum software concatenated with an auto-excitation filter wheel (Scanalytics Inc., Chicago, IL, USA) with channels for DAPI (blue), Cy3 (red), and FITC (green).

### 4.5. Immunocytochemistry and Calcium Imaging

HEK293 cells were transiently transfected using Lipofectamine 2000 (Invitrogen, Carlsbad, CA, USA) according to the manufacturer’s protocol with cDNA constructs coding for the rhesus macaque bitter taste receptor TAS2R38 [5]. The coding regions of the receptors were fused N-terminally to the sst epitope, and C-terminally to a 1D4-tag. An empty pEAK10 vector was used as a negative control. Next, transfected cells were seeded into poly-D-lysine coated chambers and incubated overnight at 37 °C in 5% CO_2_ [20]. Then, cells were washed with PBS-T, cooled on ice for 30 min, and incubated with biotin-conjugated concanavalin A (5 μg/mL, Sigma) for 1 h at 4 °C. After washing with PBS-T, the cells were fixed by treatment with methanol/acetone (1:1, *v*/*v*) for 2 min. Cells were washed again with PBS-T and incubated for 1 h with 10% normal horse serum. Primary antibodies were added at several concentrations and incubated overnight at 4 °C (Appendix A). After washing with PBS-T and blocking with 10% normal horse serum, cells were incubated with the secondary antibodies streptavidin and AMCA conjugate for 1 h at room temperature (Appendix A). Finally, cells were washed with PBS-T and fluorescence was measured by fluorescence microscopy with channels for DAPI (blue), Cy3 (red), and FITC (green) (LCM-510; Carl Zeiss Microscopy GmbH). Calcium imaging was performed to check the response of T2R38 according to the methods previously reported [5].

## Abbreviation

T2RBitter taste receptor proteinTAS2RBitter taste receptor gene in primatesTas2RBitter taste receptor gene in rodentsCVCircumvallate papillaePTCPhenylthiocarbamideDAPI4’,6-Diamidino-2-phenylindole

## Figures and Tables

**Figure 1 ijms-21-00902-f001:**
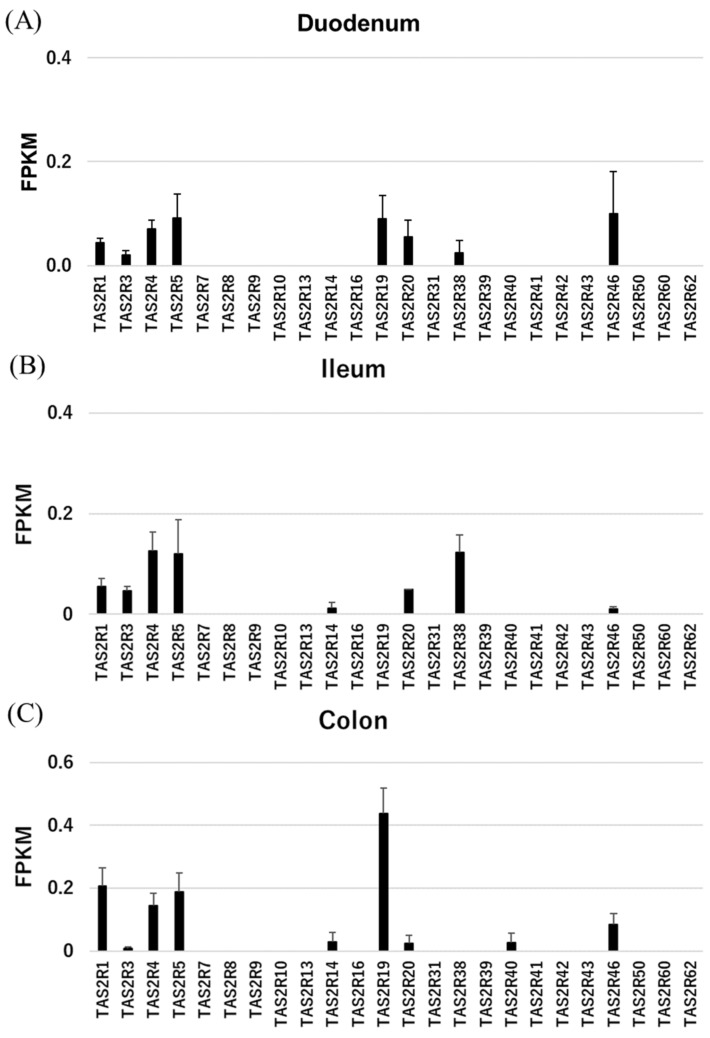
Expression levels of TAS2Rs as measured by RNAseq, which was performed using samples from the rhesus macaque duodenum (**A**), ileum (**B**), and colon (**C**). Each *n* = 3.

**Figure 2 ijms-21-00902-f002:**
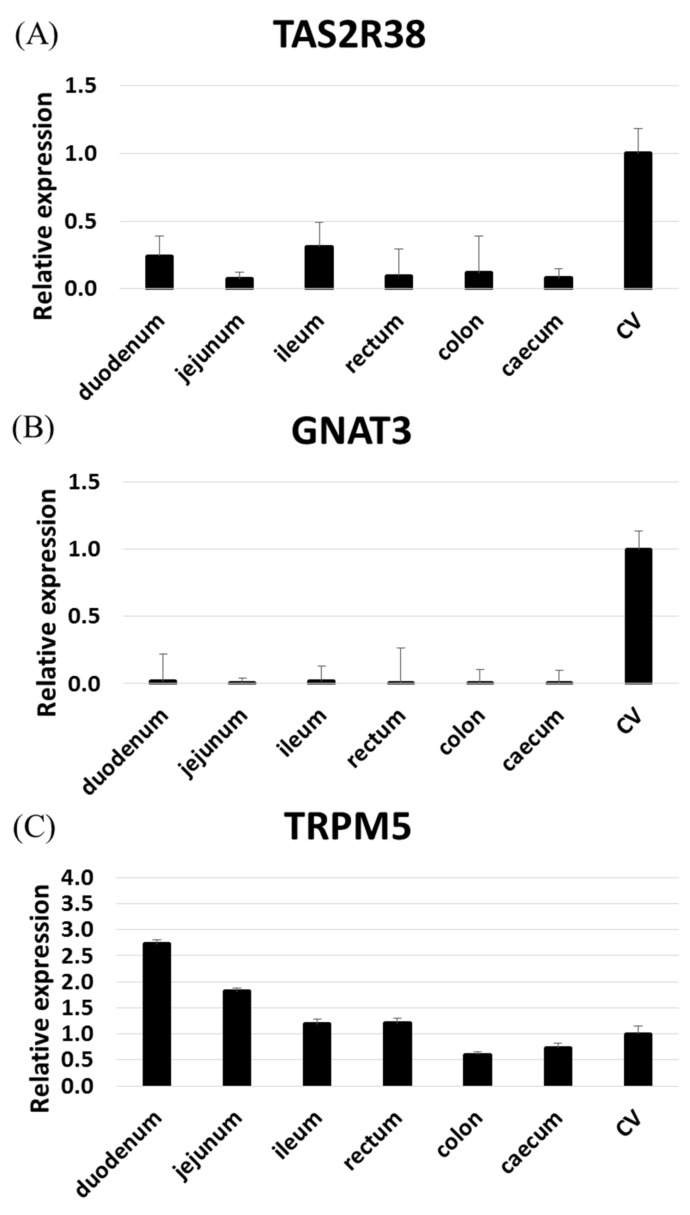
Expression levels of TAS2R38 (**A**), GNAT3 (**B**), and TRPM5 (**C**) in rhesus macaque tissue samples (*n* = 3) as determined by qRT-PCR. The expression levels of TAS2Rs were normalized against the expression in circumvallate papillae (CV) (set to 1) using GAPDH and beta-actin as reference.

**Figure 3 ijms-21-00902-f003:**
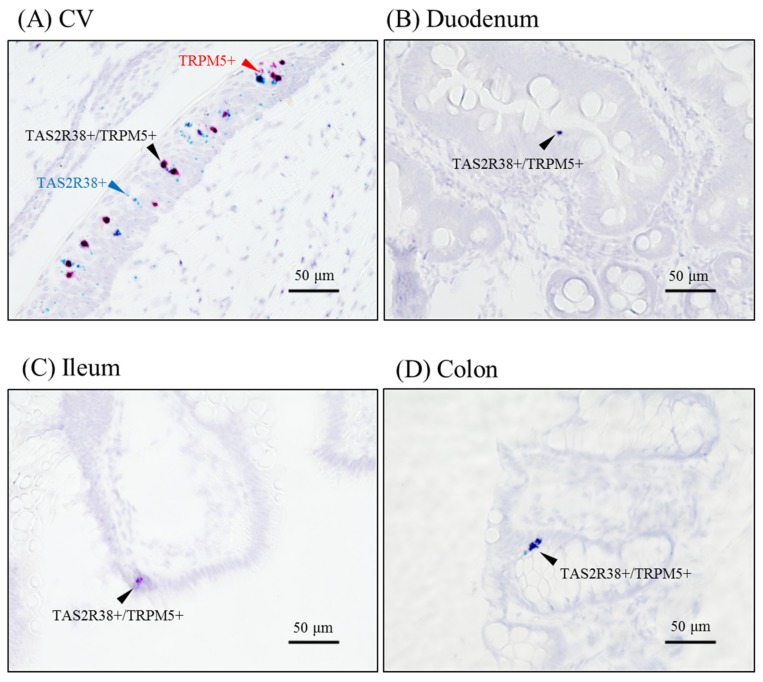
The mRNA localization in the CV and gut, as determined by RNAscope. Probes for TAS2R38 (blue) and TRPM5 (red) were hybridized and visualized in the rhesus macaque CV (**A**), duodenum (**B**), ileum (**C**), and colon (**D**). Arrowheads indicate TAS2R38+/TRPM5+ cells. Bar = 50 µm.

**Figure 4 ijms-21-00902-f004:**
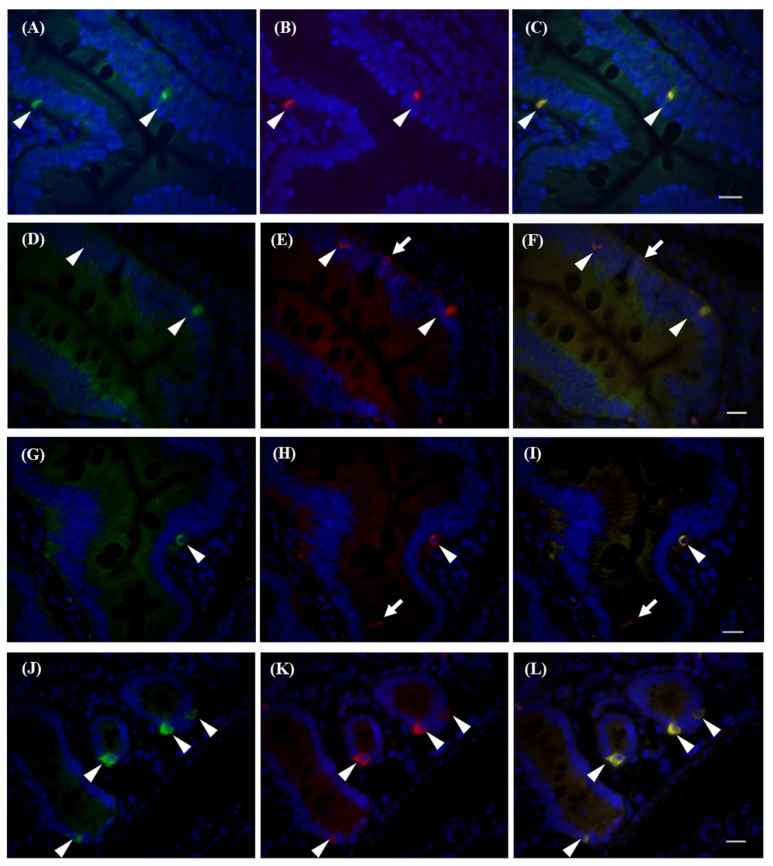
Immunohistochemical detection of T2R38-positive cells in the duodenum. T2R38-positive cells were visualized by antibodies against T2R38 (green in **A**, **D**, **G**, **J**), chromogranin A (red in **B**), Mucin2 (red in **E**), DCLK1 (red in **H**), gustducin (red in **K**), and DAPI (blue) with overlay (**C**, **F**, **I**, **L**). Arrowheads indicate T2R38+/marker+ cells. Arrows indicate T2R38−/marker+ cells. Bar = 10 µm.

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
