# Peer review of "Expression of Bitter Taste Receptors in the Intestinal Cells of Non-Human Primates"

_ijms, 2020, doi:10.3390/ijms21030902_

Round 1

Reviewer 1 Report

The here presented work describes the gene and protein expression levels of the human taste receptors type 2 in different tissues of the gastro-intestinal tract of rhesus macaques. Since at least some of the human TAS2Rs are expressed in tissues of macaques, these animals could be used as model systems in the research of bitter taste and gastro-intestinal functions of bitter compounds. I suggest accepting this article with minor revisions.

General remarks:

1 The nomenclature of bitter taste receptors should be used correctly: human genes in upper case letters (e.g. TAS2R…) and mouse or rat genes in upper and lower case letters (e.g. Tas2R…).  Proteins are named as T2R… (Annu Rev Nutr. 2007;27:389-414. Doi: 10.1146/annurev.nutr.26.061505.111329). This includes text and figures.

2 The number of biological replicates for all results should be given at least in the text, but better in figure captions. It is not enough to only state that in total samples of 8 monkeys were acquired. For instance, there is the possibility that some experiments did not work for some of the samples.

3 More details should be given on the reasoning why certain experiments were performed in that particular way (also see comments below).

Gene expression analysis:

4 Did the authors also search for non human taste receptors of type 2 in the RNAseq data? There could be the possibility that macaques have their own species-specific taste receptors. This is very important, because macaques might react differently to bitter stimuli than humans because of a differing repertoire of bitter taste receptors and this could bias results when using macaques as animal models for human bitter taste and bitter taste receptor functions.

5 Why were TAS2R38 and GNAT3 chosen for the qPCR-analysis? And why were only these two genes analyzed? Other TAS2Rs with higher expression levels would be very interesting too and could further confirm the results obtained from RNAseq.

6 According to the minimum information for publication of qPCR experiments (Clin Chem. 2009 Apr;55(4):611-22. doi: 10.1373/clinchem.2008.112797.) the normalization of qPCR results "against a single reference gene is not acceptable unless the investigators present clear evidence for the reviewers that confirms its invariant expression under the experimental conditions described". Hence, I want to suggest to use the mean of at least 2 or 3 reference genes for normalization of the qPCR results and reanalyze the qPCR data.

7 Why was TRPM5 chosen for the RNAscope experiments and not GNAT3 as in the qPCR analysis and vice versa?

8 What kind of library preparation was performed? What instrument was used for sequencing? How was the RNAseq data analyzed and processed?

9 Supplemental Table S1 is missing information on the primers. Please add GAPDH primer sequences.

10 What is the source of the primers? Were they used and published before (if yes, please cite)? Have the amplicons been sequenced to test and validate gene specificity of the primers?

Immunohistochemistry/Immunocytochemistry:

11 The presentation of the results and the discussion of immunohistochemistry should be revised:

a Clearly state which TAS2R members were analyzed in which tissues -> TAS2R14 and TAS2R38 in CV (for evaluation of antibodies), duodenum, ileum, and colon. Currently this is described very confusingly in the text.

b Why were TAS2R14 and TAS2R38 chosen for IHC and ICC labelling?

c Also the reasoning for choosing any of the mentioned markers needs to be given in the text. Which protein is used as marker for which cell type. Please, also give references for the marker proteins.

d The statement in the sentence starting in line 83 is not supported by the data given. There is no evidence for the gustducin-dependent signal transduction, because no functional analysis was performed. The results only show the gene expression of gustducin and TAS2R38 on a RNA and protein level. Hence this sentence needs to be changed, e.g.: We also detected gustducin and TAS2R38 co-expression, making gustducin-dependent signal transduction in these cells possible.

e The protein expression of TAS2R38 in the different tissues (supplementary figures S3 and S4) should be mentioned

f Sentence line 88: results for TAS2R14 are only shown in figures S5-7, not S3 and S4

12 The blue staining in all figures needs to be explained in the respective figure caption. It needs to be mentioned in the text as well and the staining protocol needs to be explained in the methods.

13 Supplemental Table S1 is missing information on the used antibodies. Against which species were the antibodies raised (human or macaques)?

14 More details on the staining protocol should be provided:

a composition of blocking solution or, if not mixed yourself, were it was bought

b washing procedure: composition of solution, how many washing steps and for how long between primary and secondary and after secondary antibody incubation

c type/model of fluorescence microscope and laser lines

d source of “normal horse serum”

e detection of concanavalin A (the protocol only states that it is biotin-conjugated, but what molecule was used for fluorescence detection and what channel was used?)

f how was “blue” staining performed, before/after/during antibody staining, what molecule was used for staining and what was the target?

Author Response

Answer to the comments of reviewer 1

Thank you for your positive comments. We revised the manuscript according to your suggestion.

1 The nomenclature of bitter taste receptors should be used correctly: human genes in upper case letters (e.g. TAS2R…) and mouse or rat genes in upper and lower case letters (e.g. Tas2R…).  Proteins are named as T2R… (Annu Rev Nutr. 2007;27:389-414. Doi: 10.1146/annurev.nutr.26.061505.111329). This includes text and figures.

Answer: Thank you for your suggestion. We checked the nomenclature in detail in text and figures.

2 The number of biological replicates for all results should be given at least in the text, but better in figure captions. It is not enough to only state that in total samples of 8 monkeys were acquired. For instance, there is the possibility that some experiments did not work for some of the samples.

Answer: Thank you for your suggestion. We added the number of biological replicates in figure captions.

3 More details should be given on the reasoning why certain experiments were performed in that particular way (also see comments below).

Answer: We added some sentences in the head of each sections.

Gene expression analysis:

4 Did the authors also search for non human taste receptors of type 2 in the RNAseq data? There could be the possibility that macaques have their own species-specific taste receptors. This is very important, because macaques might react differently to bitter stimuli than humans because of a differing repertoire of bitter taste receptors and this could bias results when using macaques as animal models for human bitter taste and bitter taste receptor functions.

Answer: We searched for non-human taste receptors of type 2 in the RNAseq data with confirming the annotation with human TAS2Rs. Unfortunately, some of TAS2Rs have quite complicated orthologues to human TAS2Rs by gene duplication, in this manuscript, we did not use the data of TAS2Rs whose orthologues are not clear between human and macaques. To address this point clearly, we added following sentence to Methods section.

P7 L160 “Analysis was performed by GeneBay, as confirming the annotation of TAS2Rs of RheMac2 as in Table S1.”

5 Why were TAS2R38 and GNAT3 chosen for the qPCR-analysis? And why were only these two genes analyzed? Other TAS2Rs with higher expression levels would be very interesting too and could further confirm the results obtained from RNAseq.

Answer: We focused on TAS2R38, because of the availability of appropriate antibody and RNAscope probes. In addition, we have a data of functionality of T2R38; human and macaque show similar response to bitter compound but mouse orthologue don’t show the response.

6 According to the minimum information for publication of qPCR experiments (Clin Chem. 2009 Apr;55(4):611-22. doi: 10.1373/clinchem.2008.112797.) the normalization of qPCR results "against a single reference gene is not acceptable unless the investigators present clear evidence for the reviewers that confirms its invariant expression under the experimental conditions described". Hence, I want to suggest to use the mean of at least 2 or 3 reference genes for normalization of the qPCR results and reanalyze the qPCR data.

Answer: Thank you for your suggestion. We added beta-actin as reference gene to qPCR and analyzed again. The results are shown in Table S3.

7 Why was TRPM5 chosen for the RNAscope experiments and not GNAT3 as in the qPCR analysis and vice versa?

Answer: We choose TRPM5 because there is commercial RNAscope probe which can be used with TAS2R38. Because we ordered the probe for GNAT3, we would like to confirm the expression of GNAT3 while we don’t have enough time to analyze within the revision deadline.

8 What kind of library preparation was performed? What instrument was used for sequencing? How was the RNAseq data analyzed and processed?

Answer: We ordered the sequencing and analysis to the specialist institute and company. Details were added as follows:

P6 L158 “RNAseq was performed at The Research Institute for Microbial Diseases, Osaka University, using TruSeq Stranded mRNA Sample Prep Kit and HiSeq2500 as sequencer. Analysis was performed by GeneBay, as confirming the annotation of TAS2Rs of RheMac2 as in Table S1.”

9 Supplemental Table S1 is missing information on the primers. Please add GAPDH primer sequences.

Answer: We are sorry for missing information. Because we analyzed again for the RT-qPCR data, we added the information of probes in new Table S2.

10 What is the source of the primers? Were they used and published before (if yes, please cite)? Have the amplicons been sequenced to test and validate gene specificity of the primers?

Answer: We used the primers previously published in the recommended paper for β-actin and GAPDH (Clin Chem. 2009 Apr;55(4):611-22. doi: 10.1373/clinchem.2008.112797.) or TAS2R38, GNAT3 and TRPM5, and confirmed the sequences before (Gonda et al., 2013). We cite these papers.

Immunohistochemistry/Immunocytochemistry:

11 The presentation of the results and the discussion of immunohistochemistry should be revised:

a Clearly state which TAS2R members were analyzed in which tissues -> TAS2R14 and TAS2R38 in CV (for evaluation of antibodies), duodenum, ileum, and colon. Currently this is described very confusingly in the text.

Answer: Because we focus on T2R38, we stated the result of T2R38 one by one.

b Why were TAS2R14 and TAS2R38 chosen for IHC and ICC labelling?

Answer: When we tried to use commercial antibodies of several TAS2Rs (listed in new table S4), only T2R14 and 38 worked with low background and appropriately labeling to heterologous expressed proteins.

c Also the reasoning for choosing any of the mentioned markers needs to be given in the text. Which protein is used as marker for which cell type. Please, also give references for the marker proteins.

Answer: We added reason and references as follows:

P4 L90 “There are several types of cells in the intestinal epithelium. In these cells, expression of T2Rs are limited to some specific types of cells: Paneth cells in the small intestine [7] and in goblet cells in the large intestine [11], enteroendocrine cells [9], and tuft cells [12]. T2R38 antibody immunoreactivity was observed in both enteroendocrine cells and goblet cells marked by chromogranin A [13] (arrowheads in Fig. 4A–C) and mucin2 [14] (arrowheads in Fig. 4D–F), respectively.”

d The statement in the sentence starting in line 83 is not supported by the data given. There is no evidence for the gustducin-dependent signal transduction, because no functional analysis was performed. The results only show the gene expression of gustducin and TAS2R38 on a RNA and protein level. Hence this sentence needs to be changed, e.g.: We also detected gustducin and TAS2R38 co-expression, making gustducin-dependent signal transduction in these cells possible.

Answer: Thank you for your point out. We revised the sentences according to your suggestion:

P4 L95 “We also detected GNAT3 and T2R38 (arrowheads in Fig. 4J-L) co-expression, making GNAT3-dependent signal transduction in these cells possible. Because we already observed macaques’ T2R38 can activate G-protein with C-terminus of GNAT3 (Galpha/gust44) [15] in the heterologous expression system [5], it is possible in the intestinal cells.”

e The protein expression of TAS2R38 in the different tissues (supplementary figures S3 and S4) should be mentioned

Answer: We mentioned as follows:

P5 L102 “Similar expression profiles are observed in ileum and colon of rhesus macaque, suggesting the difference from mouse. In case of mouse, Tas2R131 promoter worked in Paneth cells in the small intestine [7] and in goblet cells in the large intestine [11]. These results indicate that bitter taste receptors, at least T2R38, are expressed in the intestines of macaques, as in humans, whose functionality are similar to each other.”

f Sentence line 88: results for TAS2R14 are only shown in figures S5-7, not S3 and S4

Answer: Thank you for your point out. We deleted the sentence for focusing to T2R38.

12 The blue staining in all figures needs to be explained in the respective figure caption. It needs to be mentioned in the text as well and the staining protocol needs to be explained in the methods.

Answer: We mentioned in each caption for DAPI (Figure 4 and S1,S3,S4) as well as  concanavalin A (Figure S2). Staining protocol was explained in Methods section as follows:

P7 L185 “Nucleus were stained by 4',6-Diamidino-2-phenylindole (DAPI) solution (Dojindo, Japan) diluted to 1:1000 in PBS-T at room temperature for 1min.”

P7 L197  “Next, transfected cells were seeded onto poly-D-lysine coated chambers and incubated for overnight at 37℃, 5% CO2. Then, cells were washed with PBS, then cooled on ice for 30 min and incubated with biotin-conjugated concanavalin A (5μg/mL, Sigma) for 1 h at 4℃.”

13 Supplemental Table S1 is missing information on the used antibodies. Against which species were the antibodies raised (human or macaques)?

Answer: We added the information of antibodies in new Supplemental Table S4. Almost all the antibodies are raised against human proteins.

14 More details on the staining protocol should be provided:

a composition of blocking solution or, if not mixed yourself, were it was bought

b washing procedure: composition of solution, how many washing steps and for how long between primary and secondary and after secondary antibody incubation

c type/model of fluorescence microscope and laser lines

d source of “normal horse serum”

e detection of concanavalin A (the protocol only states that it is biotin-conjugated, but what molecule was used for fluorescence detection and what channel was used?)

f how was “blue” staining performed, before/after/during antibody staining, what molecule was used for staining and what was the target?

Answer: We added more details on the staining protocol in the methods section for all the comments (a-f) in 4.4. Immunohistochemistry and 4.5. Immunocytochemistry

Reviewer 2 Report

Title: Expression of bitter taste receptors in the intestinal cells of non-human primates

In this manuscript, Dr. Imai et al., studied the expression of TAS2Rs in intestinal cells of macaques. The authors report that some genes are expressed similar to humans and suggesting macaques could be good model for the gut system of humans. However, this manuscript looks preliminary to propose and establish macaques as a model system. It is required to look at other experiments that can strengthen the study before it is accepted for publication. Overall, the write up appear to be superficial and a substantial refinement is essential. Below are my concerns that need to be addressed.

1 Write up in introduction, results and discussion is inadequate. More information and details need to be provided.

2 There is no proper rational for only focusing on the expression of TAS2R38 and –R14 in follow up experiment. When TAS2R1, 3, 4, and 5 are expressed in all three tissues in RNAseq data, what is the reason for choosing TAS2R38 which is only expressed in duodenum and ilium? If the authors are interested in studying TAS2R38, the objectives need to be explained clearly.

3 Why did the authors choose TAS2R14 in immunohistochemistry? This is abrupt and no clear rational.

4 This manuscript shows only expression data. Are these expressed TAS2Rs functional? Can they recognize the same ligands as seen in humans and mice? Although the orthologs show a sequence similarity, they may differ in functionality among species. It is recommended to show the functionality of these receptors. If not all, at least selected TAS2Rs.

5 Resolution of Figure 1 is poor. It is blurred and need to provide a publication quality image.

6 In supplementary file Table S1,2, there are details that need to be filled in. For example, primer sequence details for GAPDH was left blank. In the antibodies list, details about dilution, species, code/catalogue are missing. In order to maintain consistency, I would suggest the authors to fill these gaps.

Author Response

Answer to the comments of reviewer 2

Thank you for your comments. We revised the manuscript according to your suggestion.

1 Write up in introduction, results and discussion is inadequate. More information and details need to be provided.

Answer: Thank you for your point out. We revised the sentences according to your and reviewer1’s suggestion.

2 There is no proper rational for only focusing on the expression of TAS2R38 and –R14 in follow up experiment. When TAS2R1, 3, 4, and 5 are expressed in all three tissues in RNAseq data, what is the reason for choosing TAS2R38 which is only expressed in duodenum and ilium? If the authors are interested in studying TAS2R38, the objectives need to be explained clearly.

Answer: We added the reason for focusing to T2R38 in the introduction. Briefly, there is a functional difference between primate and rodent T2R38 protein. In addition, when we tried various types of antibodies, only T2R38 and T2R14 antibodies could stain these proteins appropriately.

P1 L37 “T2R38 of human [4] and macaques [5] showed response to PTC, while mouse orthologue T2R138 showed no response to PTC [6]. In human, there is a polymorphism in the TAS2R38 gene in which PAV (taster) and AVI (non-taster) genotypes differ in the response to PTC [4]. Also, in Japanese macaque, there is a polymorphism in the TAS2R38 gene in which mutation in the second codon of starting methionine cause non-taster phenotype [5].”

3 Why did the authors choose TAS2R14 in immunohistochemistry? This is abrupt and no clear rational.

Answer: Thank you. We focused on T2R38 and deleted the sentence for T2R14 in immunohistochemistry.

4 This manuscript shows only expression data. Are these expressed TAS2Rs functional? Can they recognize the same ligands as seen in humans and mice? Although the orthologs show a sequence similarity, they may differ in functionality among species. It is recommended to show the functionality of these receptors. If not all, at least selected TAS2Rs.

Answer: As we previously reported, T2R38 of macaque is functional for PTC as seen in humans but mice T2R38 don’t respond to PTC. To make it clearer, we added the following sentence as well as introduction.

P4 L95 “We also detected GNAT3 and T2R38 (arrowheads in Fig. 4J-L) co-expression, making GNAT3-dependent signal transduction in these cells possible. Because we already observed macaques’ T2R38 can activate G-protein with C-terminus of GNAT3 (G16/gust44) [15] in the heterologous expression system [5], it is possible in the intestinal cells.”

5 Resolution of Figure 1 is poor. It is blurred and need to provide a publication quality image.

Answer: Thank you for your point out. We revised the Figure 1 according to your suggestion:

6 In supplementary file Table S1,2, there are details that need to be filled in. For example, primer sequence details for GAPDH was left blank. In the antibodies list, details about dilution, species, code/catalogue are missing. In order to maintain consistency, I would suggest the authors to fill these gaps.

Answer: We added the information to new Table S2 (primers) and S4 (antibodies)

Round 2

Reviewer 2 Report

The authors have done substantial changes to the manuscript and it improved the content. However, with only expression data, the article still appears weak.

The manuscript need to undergo extensive editing for the language used throughout the paper. I suggest the authors to get assistance from a professional editor to improvise the quality.

Once this is fulfilled, manuscript can be considered for acceptance.

Author Response

Answer to the comments of reviewer 2

Thank for your constructive comments.

We revised the manuscript according to your comments.

Comment 1:

The authors have done substantial changes to the manuscript and it improved the content. However, with only expression data, the article still appears weak.

Answer:

We added the functional response data of macaque T2R38 with human T2R38 in the same condition (Figure S2D). The result indicated the functional similarity of macaque T2R38 to human T2R38.

Comment 2:

The manuscript need to undergo extensive editing for the language used throughout the paper. I suggest the authors to get assistance from a professional editor to improvise the quality.

Answer:

We ordered specialist’ company for English editing and found extensively edited. We hope the improvement of quality.